# The Utility of 18FDG-PET/CT in Diagnosing Fever of Unknown Origin: The Experience of a Large Tertiary Medical Center

**DOI:** 10.3390/ijerph18105360

**Published:** 2021-05-18

**Authors:** Hussein Mahajna, Keren Vaknin, Jennifer Ben Shimol, Abdulla Watad, Arsalan Abu-Much, Naim Mahroum, Ora Shovman, Yehuda Shoenfeld, Howard Amital, Tima Davidson

**Affiliations:** 1Department of Medicine ‘B’, Chaim Sheba Medical Center, Tel Hashomer, Ramat Gan 52621, Israel; hmahajna@gmail.com (H.M.); kerenvaknin@gmail.com (K.V.); watad.abdulla@gmail.com (A.W.); arsalanabumuch1@gmail.com (A.A.-M.); naim.mahroum@gmail.com (N.M.); orashovman@walla.co.il (O.S.); howard.amital@sheba.health.gov.il (H.A.); 2Department of Gastroenterology, Chaim Sheba Medical Center, Tel Hashomer, Ramat Gan 52621, Israel; 3Sackler Faculty of Medicine, Tel Aviv University, Tel Aviv 6997801, Israel; jenniferb@wmc.gov.il (J.B.S.); yehuda.shoenfeld@sheba.health.gov.il (Y.S.); 4Department of Medicine, E. Wolfson Medical Center, Holon 5822012, Israel; 5Zabludowicz Center for Autoimmune Diseases, Chaim Sheba Medical Center, Tel Hashomer, Ramat Gan 52621, Israel; 6Leeds Institute of Rheumatic and Musculoskeletal Medicine, University of Leeds, Leeds LS2 9JT, UK; 7Ministry of Health of the Russian Federation, Sechenov First Moscow State Medical University, Moscow 119991, Russia; 8Department of Nuclear Medicine, Chaim Sheba Medical Center, Tel Hashomer, Ramat Gan 52621, Israel

**Keywords:** fever of unknown origin, FUO, positron emission tomography, 18FDG-PET/CT, nuclear imaging

## Abstract

Fever of unknown origin (FUO) poses a diagnostic challenge, and 18-fluorodexoyglucose positron emission tomography with computed tomography (18FDG-PET/CT) may identify the source. We aimed to evaluate the diagnostic yield of 18FDG-PET/CT in the work-up of FUO. The records of patients admitted to Sheba Medical Center between January 2013 and January 2018 who underwent 18FDG-PET/CT for the evaluation of FUO were reviewed. Following examination of available medical test results, 18FDG-PET/CT findings were assessed to determine whether lesions identified proved diagnostic. Of 225 patients who underwent 18FDG-PET/CT for FUO work-up, 128 (57%) met inclusion criteria. Eighty (62.5%) were males; mean age was 59 ± 20.3 (range: 18–93). A final diagnosis was made in 95 (74%) patients. Of the 128 18FDG-PET/CT tests conducted for the workup of FUO, 61 (48%) were true positive, 26 (20%) false positive, 26 (20%) true negative, and 15 (12%) false negative. In a multivariate analysis, weight loss and anemia were independently associated with having a contributary results of 18FDG-PET/CT. The test yielded a sensitivity of 70%, specificity of 37%, positive predictive value of 70%, and negative predictive value of 37%. 18FDG-PET/CT is a valuable tool in the diagnostic workup of FUO. It proved effective in diagnosing almost half the patients, especially in those with anemia and weight loss.

## 1. Introduction

Fever of unknown origin (FUO) is a frequently encountered condition on internal medicine wards. It was first defined in 1961 as a fever greater than 38.3 °C (101 °F) for at least 3 weeks, whose origin is not identified despite thorough work-up [1]. Despite the abundance of medical tests available to aid in the diagnosis, FUO often poses a great challenge to clinicians, and nearly half of patients ultimately do not receive a definitive diagnosis [2,3,4].

In the last decade, positron emission tomography with computed tomography (18FDG-PET/CT) has garnered widespread use outside the field of oncology because of its unique ability to survey the whole body, identifying regions of abnormal metabolism. Following the first published report of increased 18FDG enhancement in an abdominal abscess, the utility of 18FDG-PET/CT in the management of inflammatory and infectious diseases has undergone extensive exploration [5]. Utilization of 18FDG-PET/CT in infectious and inflammatory diseases is based on the uptake of FDG by granulocytes following activation, as part of a metabolic burst which is mediated by their abundant glucose transporters [6,7].

Several prospective and retrospective studies have investigated its use in the work-up of FUO and have found it highly beneficial [8,9,10,11]. Nonetheless, the literature remains with much heterogeneity regarding the place of 18FDG-PET/CT within the algorithm of recommended studies. Some consider 18FDG-PET/CT a mandatory part of the work-up, while others deem it a complementary test that may be performed following the use of other imaging modalities such as CT scans. Differences in study methodologies have produced variable results surrounding the value of PET.

With the emergence of publications supporting the use of 18FDG-PET/CT in the work-up of FUO, further studies established its fixed position within the diagnostic algorithm. A number of studies have also demonstrated that including PET/CT within the FUO algorithm is a cost-effective measure because of its diagnostic yield [12,13].

Because of the diversity surrounding this topic within the published literature, our objective was to evaluate the patients from our institution presenting with FUO. In the current study, the utility of 18FDG-PET/CT in revealing the correct cause of FUO was assessed and parameters that may increase the likelihood of diagnostic 18FDG-PET/CT in patients with FUO were sought.

## 2. Materials and Methods

### 2.1. Study Design

The medical records of all patients who underwent 18FDG-PET/CT scans as part of FUO work-up between 1 January 2013 and 30 January 2018 at the Sheba Medical Center (Ramat Gan, Israel) were reviewed. With more than 1900 beds, this is the largest and most comprehensive tertiary medical center in the Middle East. Imaging data was obtained from the picture archive and communication system (PACS, Carestream Health 11.0, Rochester, NY, USA) and clinical data were obtained from the computerized medical records within our medical center.

### 2.2. Participants

Inclusion criteria were: 1. adults 18 years old or greater; 2. who met a diagnosis of FUO according to the classic Petersdorf’s criteria, body temperature > 38.3 °C on several occasions for at least 3 weeks; 3. with no diagnosis despite one week of medical work-up; 4. and underwent 18FDG-PET/CT to search for a source of fever. The study population was ethnically heterogeneous, mainly consisting of Jews from diverse descent and a minority of Arabs. Study exclusion criteria were: pregnancy, insufficient data from medical records, active solid or hematologic malignancy, neutropenia, nosocomial infections, and HIV carrying status.

### 2.3. Medical Work-Up

Work-up prior to 18FDG-PET/CT universally included a thorough history with questioning related to fever episodes, associated symptoms, and weight loss, defined as an unintentional loss of at least 5% of body weight over the prior six months; physical examination; a complete blood count, peripheral blood smear, blood chemistry, blood cultures, C-reactive protein (CRP), erythrocyte sedimentation rate, protein electrophoresis, urinalysis, urine cultures, at least three blood cultures drawn on different occasions and from different sites; autoimmune serologies anti-nuclear antibodies, anti-neutrophil cytoplasmic antibodies, rheumatoid factor, anti-streptolysin O; varying infectious diseases serologies including those for human immunodeficiency virus (HIV), Epstein Barr virus, cytomegalovirus, and Coxiella burnetti; hepatitis B and C serologies in the instance of abnormal liver tests; chest radiograph; abdominal sonography or chest and abdominopelvic CT based on the decision of the providing clinician considering age, body habitus, renal function, and the index of suspicion that the source of fever was accessible by ultrasound particularly along the biliary tract, liver, spleen, and urinary tract; and transthoracic echocardiography.

### 2.4. 18FDG-PET/CT Imaging Technique

18FDG-PET/CT examinations were performed with a combined 18FDG-PET/CT scanner (Philips Gemini GXL, Philips Medical Systems, Cleveland, OH, USA) which includes a 16-detector row helical CT. This enables simultaneous acquisition of up to 45 trans axial PET images with inter-slice spacing of 4 mm in one bed position; and provides an image from the vertex to the thigh in most instances, with about 10 bed positions. In those patients with clinical or suspected findings in the regions below the mid-thigh according to referral, whole bodies were scanned. The trans-axial field of view and pixel size of the PET images reconstructed for fusion were 57.6 cm and 4 mm, respectively, with a matrix size of 144 × 144. The technical parameters used for CT imaging were pitch 0.8, gantry rotation speed 0.5 s/rotation, 120 kVp, 250 mAs, 3 mm slice thickness, and specific breath-holding instructions [14,15,16].

After 4–6 h of fasting, patients received an injection of an average dose of 370 MBq-100mCi F-18FDG (adapted to body weight: 0.14 mCi per 1kg of the weight of the patient). All patients with clinical questions about the heart, including suspected endocarditis, used a “cardiac regimen” to lessen physiological cardiac uptake, with a diet enriched in fat and lacking in carbohydrates for 12–24 h prior to the scan followed by a prolonged fast. About 75 min after an intravenous injection of radioactive material, CT images were obtained from the vertex to the mid-thigh for about 32 s. When intravenous contrast material was used, CT scans were obtained 60 s after injection of 2 mL/kg of non-ionic contrast material (Omnipaque 300; GE Healthcare, Chicago, IL, USA). An emission PET scan followed in 3D acquisition mode for the same longitudinal coverage, 1.5 min per bed position. CT studies were performed on the following CT scanners: Mx8000 Quad 4-MDCT, Mx8000 IDT 16-MDCT, and Brilliance 40-MDCT, 64-MDCT, and 128-MDCT (all Philips Healthcare) and 64-MDCT VCT Light-Speed (GE Healthcare). Slice thickness was reconstructed in bone and soft-tissue algorithms, reformatted in multiple planes, and evaluated in the axial orientation. CT images were fused with the PET data and were used to generate a map for attenuation correction. PET images were reconstructed using a line of response protocol with CT attenuation correction, and the reconstructed images were generated for review on a computer workstation (Extended Brilliance Workstation, Philips Medical Systems, Cleveland OH, US [14,15,16].

### 2.5. Image Assessment

All of the scans identified in the current study were reviewed by an experienced physician with two specializations (nuclear medicine and radiology, TD), with 20 years’ experience. FDG uptake in the lesions was measured by standardized uptake values max (SUVmax), which was calculated by manually generating a region of interest over the sites of abnormally increased FDG activity [14,15,16].

### 2.6. Data Collection and Review of 18FDG-PET/CT Contribution

For each patient, data was collected on their demographics, medical history, current episode of fever including pattern and duration, as well as associated signs and symptoms, and laboratory tests. 18FDG-PET/CT studies were examined and evaluated for determination of whether the test led to a final diagnosis within three months of presentation. Timing was agreed upon based on prior similar studies and the consensus that all associated symptoms generally manifested within three months and that it was a reasonable length of time for a comprehensive work-up [11,17]. Contributiveness of 18FDG-PET/CT was decided by two internal medicine physicians. When controversies arose regarding the diagnostic value of a test, the data was reviewed by a third internal medicine physician, and a conclusion was reached based on the rule of the majority.

Tests were considered true positives and therefore contributory if they had positive findings that led to the final diagnosis. Studies were labelled as true negative if no positive hypermetabolic findings were observed, and no clinical diagnosis was established during three month follow-up. True negative tests, as defined by a negative 18FDG-PET/CT scan, with no diagnosis made within 3 months, were not considered contributory. Tests were called false positive if 18FDG-PET/CT showed hypermetabolic findings, but these findings did not add to the final diagnosis; or, if no diagnosis was made at all within 3 months of follow-up. They were considered false negative if no hypermetabolic findings were observed and the diagnosis was made by other means of investigation.

### 2.7. Statistical Analysis

Descriptive statistics were used to characterize the basic features of the data in the study. The features of patients who had contributory tests were compared with those who had non-contributory ones. The Chi-square test and Fisher’s exact test were used for categorical variables, and the T-test and Mann Whitney U test for continuous variables, as appropriate. A *p*-value of less than 0.05 was considered significant in all the tests. Parameters found to be significantly associated with contributory results of 18FDG-PET/CT (true positive) on univariate analysis were analyzed in a multivariate analysis using forward step logistic regression analysis.

Statistical analysis was done using IBM SPSS for Windows, Version 23.0, Armonk, NY, USA: IBM Corp.

### 2.8. Ethical Approval

This single-institution study was approved by the institutional review board of Sheba Medical Center (SMC-18-4875). Informed consent was waived by the institutional review board due to the retrospective nature of the study. All methods were performed in accordance with the institution’s guidelines and regulations.

## 3. Results

### 3.1. Study Population

In total, 225 patients were identified as having undergone 18FDG-PET/CT scans as part of their work-up for FUO. Of these, 97 were excluded: 12 for insufficient data, seven were pediatric patients, 66 did not meet FUO criteria, one was diagnosed prior to 18FDG-PET/CT scan, and 11 had active malignancies. Among the 128 patients who met eligibility criteria and were included within the cohort, 80 (62.5%) were males. The mean age of the patients was 59 ± 20.3 years (range: 18–93). The mean duration of fever (prior to the performance of 18FDG-PET/CT) was 38 ± 34.4 days (range 14–180). Twenty-nine (22.8%) of the patients were considered immunosuppressed due to concomitant treatment with corticosteroids used as monotherapy or with immunosuppressive medications (n = 19), or immunosuppressive agents alone. Twenty-two (17%) patients had implanted prosthetic devices, including vascular implants, pacemakers and heart valves. Table 1 presents the demographic and clinical characteristics of the study population, classified according to whether 18FDG-PET/CT provided contributory results.

### 3.2. Final Diagnoses

By the completion of work-up, ninety-five patients had (74.2%) received a final diagnosis. The largest portion of patients, sixty-one (64.2% of the diagnosed patients and 47.7% of the study population) were diagnosed with infectious diseases. These most frequently consisted of endovascular infections in 14 (23.0% of infections) (Figure 1, Figure 2 and Figure 3).

Twenty-one (16.4%) of the patients were diagnosed with inflammatory diseases including inflammatory arthritis, including familial Mediterranean fever (FMF), adult onset Still’s disease (AOSD), and large- and medium-vessel vasculitides (Figure 4, Figure 5 and Figure 6). Malignancies were diagnosed in 12 patients (9.4%), nine (75%) of which were hematologic. One patient (0.8%) was diagnosed with Kikuchi disease. For 33 (25.8%) patients, no diagnosis was reached at 3 months following admission.

### 3.3. Contribution of 18FDG-PET/CT to FUO Workup

Overall, eighty-six (67.2%) of the 128 included patients had positive findings on 18FDG-PET/CT scans. In 61 (47.7%) patients, the results were deemed true positive for leading to the final diagnosis. The remaining 26 (20.3%) had false positive results. Among those with false positives, a final diagnosis was found in 17 (65.4%), and infectious causes in 13, of which six were endovascular in origin. Inflammatory sources were ultimately identified in three cases and lymphoma in the last patient (Figure 7). The most common 18FDG-PET/CT findings which were deemed false positive included increased uptake in the bone marrow, spleen, and liver, and retroperitoneal lymphadenopathy (Table 2).

The characteristics of 18FDG-PET/CT results in patients in whom a diagnosis was established by the end of the investigation.

Forty-two (33%) of the included patients did not have pathological FDG uptake findings on 18FDG-PET/CT. In twenty-six (20.3% of the study population and 61.9%% of patients with negative 18FDG-PET/CT), a final diagnosis was not achieved at the end of the workup or at three-month follow-up and were thus considered true negative. The remaining 15 (11.7%) patients without a pathological FDG uptake on 18FDG-PET/CT were eventually diagnosed using other modalities and were therefore considered false negative. Among these patients, 11 (73.3%) were found to have infections, most commonly viral, and endovascular infections, each found in three. Other cases consisted of pneumonia, pyelonephritis, cholangitis, bacteremia, and Q fever. The remaining four (26.7%) patients had inflammatory sources of fever, composed of AOSD in two, FMF, and polymyalgia rheumatica.

With regard to 18FDG-PET/CT’s ability to diagnose the cause of FUO, our findings yielded a test sensitivity of 70.1% (95% confidence interval (CI): 0.60–0.79), specificity of 36.6% (95% CI: 0.24–0.52), positive predictive value (PPV) of 70.1% (95% CI: 0.60–0.79), and negative predictive value (NPV) of 36.6% (95% CI: 0.24–0.52).

### 3.4. 18FDG-PET/CT Contributiveness According to Disease Category

Of patients with final diagnoses of infectious and inflammatory conditions, the rates of true positive results were 61.7% and 60%, respectively, and the rates of false negative results were 20.0% and 30.0%, respectively. Of the patients who were given a final diagnosis of malignancy, 91.7% had true positive 18FDG-PET/CT findings and 8.3% of them had false positive results. There were no false negative results.

### 3.5. Role of Iatrogenic Immunosuppression

Out of the 61 patients found to have true positive 18FDG-PET/CT results, 17 (28%) were receiving immunosuppressive treatment. Of these 17, five (29%) were taking corticosteroids alone, five (29%) were being treated with combination corticosteroid and immunosuppressive medications, and seven (41%) were using immunosuppressants exclusively. Six (25%) of the patients whose 18FDG-PET/CT studies were considered true negatives were medically immunosuppressed. Three (12.5%) of the 25 found to be false positive were receiving immunosuppressive therapies: two on corticosteroid monotherapy and one with combination corticosteroids and an additional immunosuppressant. Three (20%) of the 18 false negative tests were found in patients being treated with immunosuppression: one with corticosteroids combined with an additional immunosuppressive agent, and two with non-corticosteroid immunosuppressing treatments alone.

### 3.6. 18FDG-PET/CT Contributiveness in Those Who Underwent Prior CT Imaging

Fifty-eight (45.3%) of the patients underwent abdominal, pelvic and thoracic CT scans, with use of intravenous contrast media unless contraindicated, prior to performance of 18FDG-PET/CT, based on the judgment of providing clinicians. Of these fifty-eight, findings were found to be true positive in 27 (46.5%) and true negative in 13 (22.4%). False positive findings were detected in 14 (24.1%). In four (6.9%), the absence of 18FDG-PET/CT was determined to be false negatives. Statistical analysis revealed a sensitivity of 65.9% (95% CI: 0.51–0.78), specificity 23.5% (95% CI: 0.96–0.47), PPV 67.5% (95% CI: 0.52–0.80), and NPV 22.2% (95% CI: 0.24–0.52) for 18FDG-PET/CT in diagnosing the source of fever in those who had undergone prior CT scanning.

### 3.7. Parameters Associated with True Positive Results

In a univariate analysis, weight loss, as assessed retrospectively by subjective questioning of the patients, low concentrations of hemoglobin and low levels of transferrin were associated with true positive diagnoses. In a multivariate analysis, level of hemoglobin was inversely associated with contributory 18FDG-PET/CT tests (odds ratio (OR): 0.597, 95% CI 0.412–0.866, *p* = 0.006). Weight loss, on the other hand, was positively associated with contributory 18FDG-PET/CT tests (OR: 3.605, 95% CI 1.123–11.576, *p* = 0.031). The association of low-level transferrin with a diagnostic 18FDG-PET/CT was not significant: (OR: 0.992, 95% CI 0.981–1.003, *p* = 0.13.

## 4. Discussion

In this retrospective study, the 18FDG-PET/CT scans of a large group of patients presenting with FUO were assessed for diagnostic utility. Clinical and laboratory variables in which the diagnostic yield of 18FDG-PET/CT was highest were also identified. Our study adhered to rigorous methodology, with extensive medical work-up, including CT scans of the chest, abdomen and pelvis in close to half of the cohort, prior to 18FDG-PET/CT. Imaging protocol was modified as appropriate, with extension of scanning fields to the region distal to the thighs in cases of suspected septic embolism, mycotic aneurisms or vasculitis of the peripheral vessels, according to previously published standards [18]. Though historically 18FDG-PET/CT has been used mostly for the diagnosis of primary tumors and the search for distal metastases [19], our findings support the growing indications for surveying the entire body for areas of increased metabolism.

18FDG-PET/CT studies were found to be diagnostically useful in almost half of these patients. A review of the current literature suggests similar findings, though the percentage of examinations in which 18FDG-PET/CT served as beneficial parts of the work-up varies widely, with numbers ranging from 42% to 72% [8,9,10,11,17,20,21,22,23,24,25,26,27,28]. This variation can be explained, in part, by differences in the definitions of FUO, diagnostic algorithms and the place of 18FDG-PET/CT within the algorithms, technique and model of the 18FDG-PET/CT scanners, and by how usefulness of the 18FDG-PET/CT is defined [29]. Moreover, studies differed in whether they included immunocompromised patients and the extent to which immunosuppressive medications were used, widening the discrepancies [30,31].

In our study, just under half of the 18FDG-PET/CT scans performed were found to be true positives, corresponding to the typical rate published in the literature. In spite of the wide range of methodologies and the frequent use of small sample sizes, most studies show that 18FDG-PET/CT is diagnostic in 50%–60% of patients with FUO ([20,25,27,28]). One retrospective study from South Korea found true positive results in 52.1% of patients [22]. Another demonstrated 18FDG-PET/CT useful in diagnosis in 56.7% of the patients with FUO. However, in this study, all patients admitted for FUO work-up underwent 18FDG-PET/CT [11]. In contrast, a higher yield of 72% was found when 18FDG-PET/CT was performed only following the completion of a first-line of investigation [23]. A meta-analysis of 22 studies illustrated an overall true positive rate of approximately 60% [32]. Overall, 18FDG-PET/CT is generally more contributive in studies reporting higher proportions of infections and neoplasms [32].

Despite the heterogeneity of published findings, our rate of false negative results of 18FDG-PET/CT scans, at just over 10%, is comparable to rates reported within the literature. Two thirds of these patients were diagnosed with infectious diseases and one third with inflammatory diseases while none were diagnosed with malignant diseases. Similarly, in an earlier study, Gafter-Gvili et al. [10] reported a rate of false negatives of 18%; of these, 40% were eventually diagnosed with infectious disease and only one patient was diagnosed with a cancer. As prior studies have indicated, 18FDG-PET/CT plays a fundamental role in detecting occult primary malignant lesions and metastases, including lymph node involvement, even in the absence of node enlargement [33].

False positive results were found in one fifth of our patients. Of these, three quarters were found to have infections, close to one fifth were found to have inflammatory conditions, and one patient was ultimately diagnosed with lymphoma. In the remaining examinations that were found to be false positives, which composed just over two-fifths of this group, no diagnosis was made at the end of the work-up or at 3 months follow-up. Our false positive rates match those reported by other studies which range between 9% and 30% of 18FDG-PET/CT scans performed as part of FUO work-up [3,10,34,35].

Infectious etiology was the most common cause for FUO, found in close to half of patients, followed by inflammatory diseases which were identified in close to two fifths, matching the results of numerous prior publications [1,2,10,29,36]. 18FDG-PET/CT was diagnostic in just over three fifths of the patients with infections, in exactly 60% of those with inflammatory diseases, and in more than 90% of malignant causes of FUO. Cancers were mostly hematologic, making up almost three quarters of the malignancies diagnosed. 18FDG-PET/CT missed the diagnosis of cancer in only one patient, whose scan showed increase uptake in the lungs though ultimately was diagnosed with T-cell lymphoma through bone marrow biopsy. Accordingly, 18FDG-PET/CT demonstrated the highest yield in diagnosing malignant causes of FUO supporting its well established role in detecting cancerous lesions.

Immunosuppression was also assessed for the role it played in the diagnostic ability of 18FDG-PET/CT. Among the close to one quarter of patients who received immunosuppression, 18FDG-PET/CT was considered true positive in just under 60%, true negative in one fifth, false positive in one tenth, and false negative in another one tenth. No conclusions could be drawn on the effect of immunosuppression based on the slight deviation from the findings yielded among the entire cohort. Similarly, other papers have reported use of corticosteroids in up to 27% of the study population with no conclusion drawn on its effect on the diagnostic utility of 18FDG-PET/CT [10].

Our study also found that the contributiveness of 18FDG-PET/CT did not differ in the just under half of patients who had undergone extensive prior CT imaging. The breakdown of how the test was classified by the judging clinicians and the statistical analysis of test properties were numerically comparable in those who underwent prior CT relative to the entire study cohort. Likewise, in a large study evaluating 18FDG-PET/CT in cases of FUO in which more than 70% of study participants had undergone prior chest and abdominal CT imaging, no correlation was detected between prior advanced diagnostic testing and whether 18FDG-PET/CT was found to be contributory [30].

In our study, weight loss and low hemoglobin levels were independently associated with 18FDG-PET/CT producing diagnostic findings. Likewise, Crouzet et al. [8] illustrated an association between low hemoglobin levels and true positive 18FDG-PET/CT scans. Other studies have also found other associated variables including presence of adenopathy and elevated CRP [8,9,11]. However, differences in study methodology may explain the discrepancies as illustrated by one study defining both true positives and true negatives as diagnostically useful and reporting both. short duration of fever and male gender as predictors of 18FDG-PET/CT diagnostic ability [10].

Our study has several limitations. Because of its retrospective design, the diagnostic work-up of FUO before proceeding to 18FDG-PET/CT was not entirely standardized, and therefore we could not determine whether the entire basic work-up was conducted in every single patient before proceeding to PET/CT. Patients were evaluated at multiple sites and to varying extents, resulting in a non-homogenous backdrop. Moreover, the fact that our study was based on data from a single tertiary center might have introduced a selection bias. Nevertheless, our data concerned a group of patients presenting with FUO while adhering to strict methodology and exclusion criteria. Consensus among clinicians and among nuclear physicians on defining and interpreting the findings resulted in a reliable assessment of the utility of 18FDG-PET/CT in the work-up of FUO.

## 5. Conclusions

In our study, 18FDG-PET/CT was found to be highly beneficial in the work-up of FUO. It led to diagnosis in nearly half of the patients in which a diagnosis could not be established following a thorough medical evaluation. Moreover, in patients with weight loss and anemia in particular, 18FDG-PET/CT is most likely to assist in reaching the final diagnosis. Clinicians should be aware of the clinical utility of 18FDG-PET/CT and its place in the work-up of FUO when no diagnosis can be established with the basic clinical, laboratory and imaging tools. More studies using a multi-center randomized controlled design are warranted to best define the place of 18FDG-PET/CT in the work-up of FUO.

## Figures and Tables

**Figure 1 ijerph-18-05360-f001:**
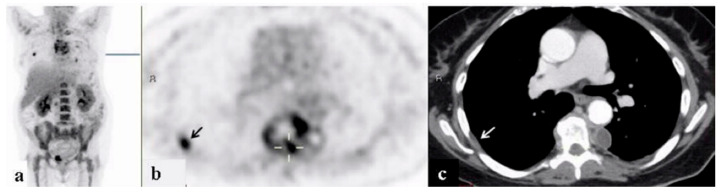
A 68-year–old woman who underwent work-up for fever of unknown origin. 18FDG-PET/CT: maximum intensity projection (MIP) (**a**) and representative PET (**b**) and CT (**c**) axial slices. PET demonstrates increased uptake in the non-homogeneous soft tissue paravertebral mass with intrathecal infiltration (curser). Additional foci of increased uptake in the pleural/extra pleural lesion appear in the inner chest wall on the right (arrow). Findings are consistent with a paravertebral abscess and septic emboli.

**Figure 2 ijerph-18-05360-f002:**
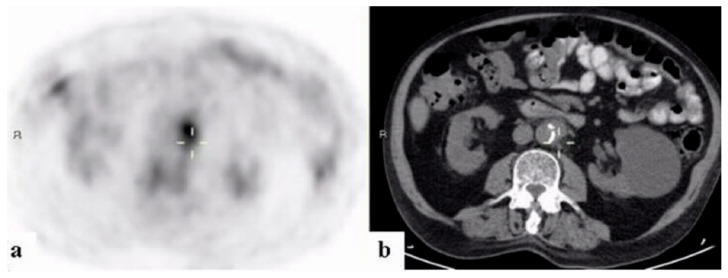
A 67-year–old man with fever of unknown origin. 18FDG-PET/CT: PET (**a**) and CT (**b**) axial slices. PET demonstrates increased uptake in the wall of the abdominal aorta corresponding with soft tissue thickening (curser). Findings are consistent with a mycotic aneurysm.

**Figure 3 ijerph-18-05360-f003:**
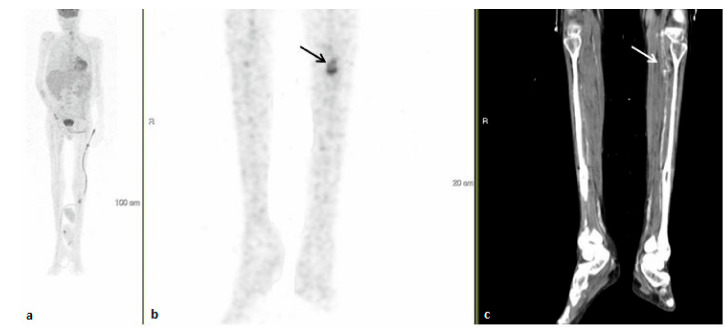
A 54-year–old man with fever of unknown origin, one month after completing treatment for endocarditis. 18FDG-PET-CT: maximum intensity projection (MIP) (**a**) and representative PET (**b**) and CT (**c**) axial slices. PET demonstrates increased uptake in the wall of the popliteal artery corresponding with aneurysmal dilatation (arrows). The findings are consistent with mycotic aneurysm.

**Figure 4 ijerph-18-05360-f004:**
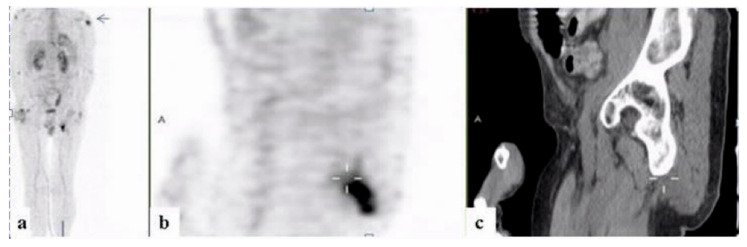
A 58-year–old man with fever of unknown origin. 18FDG-PET-CT: maximum intensity projection (MIP) (**a**) and representative PET (**b**) and CT (**c**) sagittal slices. PET demonstrates increased uptake in soft tissue thickening adjacent to the ischium (curser) and the humerus on the left (arrow), consistent with inflammation of the entheses.

**Figure 5 ijerph-18-05360-f005:**
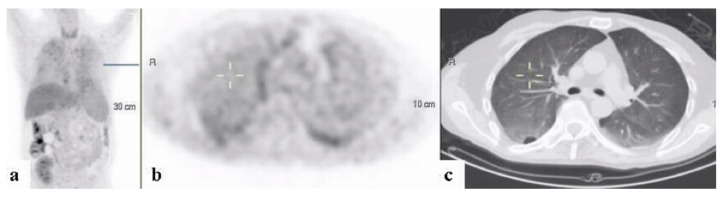
A 58-year–old man with fever of unknown origin. 18FDG-PET/CT: maximum intensity projection (MIP) (**a**) and representative PET (**b**) and CT (**c**) axial slices. PET demonstrates slightly increased uptake in the shadows with “ground glass appearance” in both lungs (curser). The findings are consistent with pneumonitis.

**Figure 6 ijerph-18-05360-f006:**
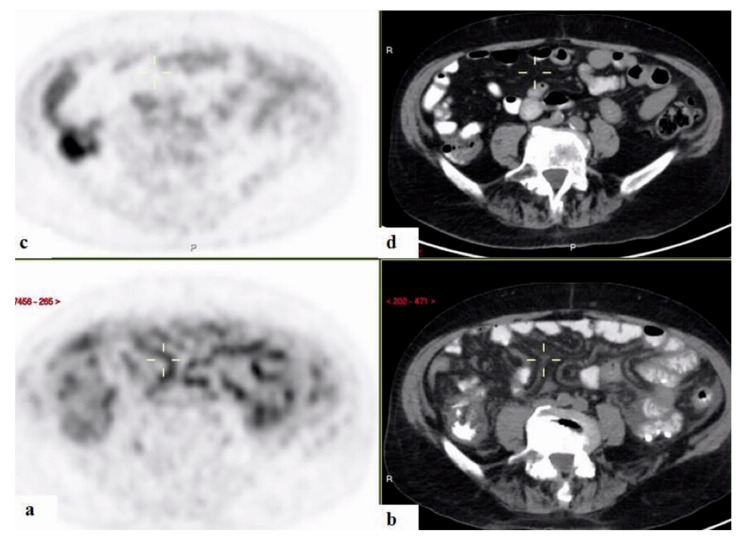
18FDG-PET/CT: PET and CT axial slices. A 71-year-old man with fever of unknown origin with biopsy proven mesenteric arteritis, performed following 18FDG-PET/CT results. PET demonstrates increased uptake along the mesenteric vessels (curser) (**a**,**b**), which disappeared following immunosuppressive treatment (**c**,**d**).

**Figure 7 ijerph-18-05360-f007:**
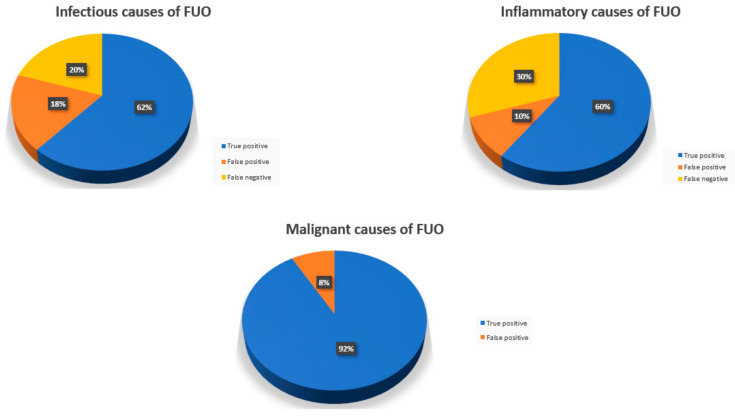
18FDG-PET/CT results according to final diagnosis disease category.

**Table 1 ijerph-18-05360-t001:** Demographic and clinical characteristics of patients with fever of unknown origin.

Variable	All, *n* = 128 (%)	Contributory PET/CT Results (*n* = 61)	Non-Contributory PET/CT Results (*n* = 67)	*p*-Value
Sex (male, %)	80 (62.5%)	63.9%	61.2%	0.75
Age (years)	59 (18–93)	56.20	61.72	0.12
Smoking	35 (27.3%)	34.4%	20.9%	0.09
BMI	25.8	25.1	26.3	0.21
Diabetes	31 (24.8%)	18.6%	30.3%	0.13
Hypertension	63 (49.2%)	44.3%	53.7%	0.28
Dyslipidemia	44 (34.4%)	27.9%	40.3%	0.14
Chronic kidney disease	16 (12.5%)	11.5%	13.4%	0.79
Congestive heart failure	14 (10.9%)	11.5%	10.4%	1.00
Vascular disease	36 (28.1%)	26.2%	29.9%	0.65
Prosthetic device	22 (17.2%)	19.7%	14.9%	0.48
Neurologic disease	21 (16.5%)	19.7%	13.6%	0.47
Previous malignancy	19 (14.8%)	18.0%	11.9%	0.46
Autoimmune disease	30 (23.4%)	29.5%	17.9%	0.12
Immunosuppressed	29 (22.8%)	28.3%	17.9%	0.16
Cortico-steroroid therapy	19 (15.0%)	16.4%	13.6%	0.80
Other immuno-suppressants	19 (15.1%)	18.0%	12.3%	0.46
Days of fever	37.9	41.0	35.2	0.34
Rash	15 (11.7%)	11.5%	11.9%	1.00
Dyspnea	14 (10.9%)	16.4%	6.0%	0.09
Weight loss	33 (25.8%)	34.4%	17.9%	0.03
Abdominal pain	24 (18.8%)	18.0%	19.4%	0.84
Diarrhea	9 (7.0%)	8.2%	6.0%	0.74
Chest pain	6 (4.7%)	3.3%	6.0%	0.68
Pharyngitis	7 (5.5%)	4.9%	6.0%	1.00
Headache	14 (10.9%)	14.8%	7.5%	0.26
Arthritis	12 (9.4%)	14.8%	4.5%	0.07
Lymphadenopathy	8 (6.3%)	6.6%	6.0%	1.00
WBC (X10^3)	9.5	9.9	9.2	0.55
HB (g/dL)	10.3	9.8	10.7	<0.01
PLT	266	274.1	259.2	0.60
Creatinine	1.27	1.16	1.37	0.33
Urea	48	46.2	49.2	0.67
AST	36.1	37.9	34.6	0.66
ALT	38.6	36.0	41.0	0.54
LDH	263.3	281.9	246.6	0.13
ALP	124.1	114.6	132.7	0.20
GGT	98.0	97.2	98.9	0.92
Albumin	3.2	3.1	3.2	0.15
CRP	95.1	104.6	86.7	0.23
ESR	73.5	80.5	68.5	0.16
ANA	16 (19.5%)	25.6%	14.0%	0.265
Ferritin	715.2	858.9	580.0	0.24
Transferrin	173.5	158.9	187.7	<0.01

ALT, alanine transaminase; ALP, Alkaline phosphatase ANA, anti-nuclear antibody; AST, aspartate transaminase; BMI, body mass index; CRP, c-reactive protein; ESR, erythrocyte sedimentation rate; GGT, gamma-glutamyl-transferase; HGB, hemoglobin; LDH, lactate dehydrogenase; PLT, platelet; WBC, white blood cell.

**Table 2 ijerph-18-05360-t002:** 18FDG-PET/CT findings and final diagnoses in examinations considered false positive in 26 patients.

Patient Number	18FDG-PET/CT Findings	Final Diagnosis
1	Lymphadenopathy	Endovascular infection
2	Increased adrenal uptake	Endovascular infection
3	PE with increased pulmonary uptake	Endovascular infection
4	Increased cecal uptake	Endovascular infection
5	Increased uptake along the sternum	Endocarditis
6	Increased uptake in the cecum and rectum	Endocarditis
7	Increased pulmonary uptake	Parvovirus infection
8	Increased BM uptake	CMV infection
9	Increased BM uptake	CMV infection
10	Increased pharyngeal and esophageal uptake	Osteomyelitis
11	Increased uptake along the R shoulder	Atypical PNA
12	Splenomegaly and increased BM uptake	Q fever
13	Increased cutaneous and subcutaneous increased uptake	Q fever
14	Increased BM uptake	AOSD
15	Increased uptake in distal ureter	Temporal arteritis
16	Phlebitis	Granulomatous hepatitis
17	Increased renal uptake	Lymphoma
18	Increased uptake of retroperitoneal LNs	No final diagnosis found
19	Increased uptake along periportal nodes	
20	Increased uptake along the ascending colon	
21	Increased uptake along the spleen, retroperitoneal LNs and in the BM	
22	Increased pulmonary and esophageal uptake	
23	Increased hepatic uptake	
24	Increased uptake along the R hip and R shoulder	
25	Increased uptake along the colon and groin	
26	Increased uptake along the retroperitoneal LNs and pelvis	

AOSD, adult-onset Still’s disease; BM, bone marrow; CMV, cytomegalovirus; LN, lymph nodes; PE, pulmonary embolism; PNA, pneumonia; Q, query; R, right.

## Data Availability

Data discussed in this article will be made readily available upon request.

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
