# Peer review of "The Utility of 18FDG-PET/CT in Diagnosing Fever of Unknown Origin: The Experience of a Large Tertiary Medical Center"

_ijerph, 2021, doi:10.3390/ijerph18105360_

Round 1
Reviewer 1 Report
Title: The utility of 18FDG-PET/CT in diagnosing fever of unknown origin: A mathematical Model
Manuscript Number: ijerph-1191267
The purpose of this study was to evaluate the utility of 18FDG-PET/CT in revealing the correct cause of FUO. Parameters that may increase the likelihood of diagnostic 18FDG-PET/CT in patients with FUO were sought.
This manuscript consists of a non-structured abstract with keywords, 5 sections (introduction, materials & methods with 8 subsections, results with 7 subsections, discussion, and conclusions) on 11 pages of single-spaced text with embedded tables and figures. There are 30 references, 2 tables, and 7 figures. No appendices. No URL is cited.
Fever of unknown origin is a challenging and important clinical condition, so evaluation of PET/CT imaging in resolving these cases is potentially very valuable. Several prior reports of PET/CT for FUO have appeared in the literature, so the principal and unique aspect of this manuscript is the single center site of origin, Sheba Medical Center (2013-2018).
The race and ethnicity of the study population are not described.
The title contains "A Mathematical Model", but this is not mentioned in the abstract or manuscript text. No "Mathematical Model" is presented and this phrase should be deleted from the title. Several sections have numerical designations that deserve review (e.g., 2.418, 3.418, 3.618) by a qualified editor.
Many prior reports of PET/CT applied to FUO are cited in the bibliography (e.g., references #16 - 25), but a detailed summary of the results from these studies is not available, and very little information is provided to compare the current and prior studies. This is a MAJOR omission. Without a detailed summary from numerous prior reports and itemized, categorial comparison between studies, it is very difficult to understand what, if any, new findings are available from the current study.
In fact, it seems that the contribution of PET/CT to FUO would be considered to be relatively minor based on "sensitivity of 70%, specificity of 37%, positive predictive value of 70%, and negative predictive value of 37%" reported in the abstract. Without a detailed comparison with previous studies of PET/CT applied to FUO and in the absence of a comparison of PET/CT vs. other modalities in FUO, it is very difficult to draw meaningful conclusions from the current study.
Selection bias is a major concern in a single center non-randomized retrospective study. Of 225 eligible patients, only 128 (57%) were included. How is this justifiable?
Multivariate statistical analysis was performed using unknown statistical software. The tools used for this analysis should be provided. The data entered into the statistical analysis are not available for independent review. "forward step regression analysis" was done. What tool was used for this analysis and what specific criteria were applied?
Since the sample size is quite small (n=228 of which only 128 met inclusion criteria), sample size justification for multivariate analysis is needed.
Numerous univariate tests were done, reported in Table 1. It is not clear that any correction was done for potential spurious significances.
Several terms are used that uncommon in reports of diagnostic imaging performance. For example, "predictiveness" in the caption of Figure 7, "contributiveness" in line 137 on page 3, and many other instances. There may be better alternatives, or the authors could provide a rigorous statistical definition of their meaning in the context of this report.
Several study limitations are presented in the last paragraph of the Discussion section, but others are present. No specific recommendations are presented on how to conduct a definitive study.
Is the definition of FUO in reference #1 entirely satisfactory or should some refinement be added based on this study?
The only graphical summary of this study is presented in Figure 7. A study schema and flowchart or timelines are absent. The addition of these graphics could increase the impact of this report. The diagnostic algorithm developed at the Sheba Medical Center for FUO management could be introduced as a graphic. This could be a major contribution due to the reputation of the institution and long experience with this condition. Issues such as criteria for selection of imaging modality and when studies should be done or repeated are especially important.
Other relevant studies of FUO and PET/CT imaging could be cited and discussed including:
Fever of Unknown Origin: the Value of FDG-PET/CT.
Kouijzer IJE, Mulders-Manders CM, Bleeker-Rovers CP, Oyen WJG.
Semin Nucl Med. 2018 Mar;48(2):100-107. doi: 10.1053/j.semnuclmed.2017.11.004. Epub 2017 Dec
FDG-PET/CT in Fever of Unknown Origin, Bacteremia, and Febrile Neutropenia.
Hess S.
PET Clin. 2020 Apr;15(2):175-185. doi: 10.1016/j.cpet.2019.11.002.
Contribution of 18F-FDG PET/CT in a case-mix of fever of unknown origin and inflammation of unknown origin: a meta-analysis.
Kan Y, Wang W, Liu J, Yang J, Wang Z.
Acta Radiol. 2019 Jun;60(6):716-725. doi: 10.1177/0284185118799512. Epub 2018 Sep 11.
PMID: 30205705
----
Overall, this summary of a single institution's experience with imaging for FUO provides an important summary from a major medical center. There is no "mathematical model", so the title must be changed. Sample size justification should be added. A detailed tabulated comparison of prior published reports and the current study should be added. Several illustrations are suggested for addition, especially the Sheba Medical Center diagnostic workup protocol, study schema, and imaging modality selection criteria. Some minor clarifications or modifications of terminology would be helpful.
Author Response
Title: The utility of 18FDG-PET/CT in diagnosing fever of unknown origin: A mathematical Model
Manuscript Number: ijerph-1191267
The purpose of this study was to evaluate the utility of 18FDG-PET/CT in revealing the correct cause of FUO. Parameters that may increase the likelihood of diagnostic 18FDG-PET/CT in patients with FUO were sought.
This manuscript consists of a non-structured abstract with keywords, 5 sections (introduction, materials & methods with 8 subsections, results with 7 subsections, discussion, and conclusions) on 11 pages of single-spaced text with embedded tables and figures. There are 30 references, 2 tables, and 7 figures. No appendices. No URL is cited.
Fever of unknown origin is a challenging and important clinical condition, so evaluation of PET/CT imaging in resolving these cases is potentially very valuable. Several prior reports of PET/CT for FUO have appeared in the literature, so the principal and unique aspect of this manuscript is the single center site of origin, Sheba Medical Center (2013-2018).
The race and ethnicity of the study population are not described.
A sentence about the study population’s ethnicity was added to the participants’ section.
The title contains "A Mathematical Model", but this is not mentioned in the abstract or manuscript text. No "Mathematical Model" is presented and this phrase should be deleted from the title. Several sections have numerical designations that deserve review (e.g., 2.418, 3.418, 3.618) by a qualified editor.
The title was amended to contain “the experience of a large tertiary medical center” instead of “a mathematical model”. The error in the numerical designation was reviewed and corrected.
Many prior reports of PET/CT applied to FUO are cited in the bibliography (e.g., references #16 - 25), but a detailed summary of the results from these studies is not available, and very little information is provided to compare the current and prior studies. This is a MAJOR omission. Without a detailed summary from numerous prior reports and itemized, categorial comparison between studies, it is very difficult to understand what, if any, new findings are available from the current study.
In fact, it seems that the contribution of PET/CT to FUO would be considered to be relatively minor based on "sensitivity of 70%, specificity of 37%, positive predictive value of 70%, and negative predictive value of 37%" reported in the abstract. Without a detailed comparison with previous studies of PET/CT applied to FUO and in the absence of a comparison of PET/CT vs. other modalities in FUO, it is very difficult to draw meaningful conclusions from the current study.
Thank you for this comment. An explanation of this issue was added to the discussion, in line 332.
Selection bias is a major concern in a single center non-randomized retrospective study. Of 225 eligible patients, only 128 (57%) were included. How is this justifiable?
We agree with the possibility of a selection bias. This was added to the limitation paragraph. Regarding the relatively low percentage of participants which were included, a detailed description of the exclusion process that led to excluding 97 participants is described in line 97 “Of them, 97 were excluded: 12 for insufficient data, seven were pediatric, 66 did not meet FUO criteria, one was diagnosed prior to 18FDG-PET/CT scan, and 11 had active malignancies.”. The high percentage of exclusion represents our effort to strictly adhere to the inclusion criteria with the retrospective data available. Special attention was directed toward the FUO criteria, which led to excluding 66 patients.
Multivariate statistical analysis was performed using unknown statistical software. The tools used for this analysis should be provided. The data entered into the statistical analysis are not available for independent review. "forward step regression analysis" was done. What tool was used for this analysis and what specific criteria were applied?
The logistic regression model was used in order to assess the independently associated parameters with contributive PET/CT scans. The forward step Wald procedure was computed in order to select the independent variables of contributive PET/CT.
All statistical tests were 2-sided and p < 0.05 was considered as statistically significant. Statistical analysis was done using IBM SPSS for Windows, Version 23.0, Armonk, NY: IBM Corp." The text was amended to include the missing information.
Since the sample size is quite small (n=228 of which only 128 met inclusion criteria), sample size justification for multivariate analysis is needed.
We appreciate your noting that the sample size is small for multivariate analysis. The multivariate analysis included three independent variables (levels of hemoglobin, weight loss, low-level transferrin). Due to the low number of variables included in the model, we assumed that a sample size of n>100 will be sufficient.
Numerous univariate tests were done, reported in Table 1. It is not clear that any correction was done for potential spurious significances.
We agree that a correction for potential spurious significances is valuable. However, correction for multiple tests was not performed for univariate tests because the purpose of these tests was to identify variables that were later included in the multivariate model. It should be noted that our conclusions regarding significance are based on the results of the multivariate analysis.
Several terms are used that uncommon in reports of diagnostic imaging performance. For example, "predictiveness" in the caption of Figure 7, "contributiveness" in line 137 on page 3, and many other instances. There may be better alternatives, or the authors could provide a rigorous statistical definition of their meaning in the context of this report.
We avoided the use of the term predictiveness and provided a definition of the term contributive test in the text in line 144.
Several study limitations are presented in the last paragraph of the Discussion section, but others are present. No specific recommendations are presented on how to conduct a definitive study.
The limitations paragraph was amended to include the issue of selection bias. A direction for future studies was added to the conclusion.
Is the definition of FUO in reference #1 entirely satisfactory or should some refinement be added based on this study?
Although the FUO definition in reference 1 dates back to 1961, it is still the basis for many studies similar to our own. For the inclusion criteria, we chose a definition that is widely accepted.
The only graphical summary of this study is presented in Figure 7. A study schema and flowchart or timelines are absent. The addition of these graphics could increase the impact of this report. The diagnostic algorithm developed at the Sheba Medical Center for FUO management could be introduced as a graphic. This could be a major contribution due to the reputation of the institution and long experience with this condition. Issues such as criteria for selection of imaging modality and when studies should be done or repeated are especially important.
Thank you for this comment. In section number 2.3 “medical work-up”, we described the routine work-up of patients who are admitted for FUO evaluation in our institution. Nevertheless, in the real world, unlike in controlled trials, this work-up varies. For instance, while some physicians chose to send the patients to CT scan first, others proceeded directly to PET/CT based on clinical judgement. In Sheba Medical Center, there is no mandatory algorithm for FUO work-up, and the management is based on the accepted steps in the literature along with the medical judgment of the physician.
Other relevant studies of FUO and PET/CT imaging could be cited and discussed including:
Fever of Unknown Origin: the Value of FDG-PET/CT.
Kouijzer IJE, Mulders-Manders CM, Bleeker-Rovers CP, Oyen WJG.
Semin Nucl Med. 2018 Mar;48(2):100-107. doi: 10.1053/j.semnuclmed.2017.11.004. Epub 2017 Dec
FDG-PET/CT in Fever of Unknown Origin, Bacteremia, and Febrile Neutropenia.
Hess S.
PET Clin. 2020 Apr;15(2):175-185. doi: 10.1016/j.cpet.2019.11.002.
Contribution of 18F-FDG PET/CT in a case-mix of fever of unknown origin and inflammation of unknown origin: a meta-analysis.
Kan Y, Wang W, Liu J, Yang J, Wang Z.
Acta Radiol. 2019 Jun;60(6):716-725. doi: 10.1177/0284185118799512. Epub 2018 Sep 11.
PMID: 30205705
Thank you for directing us to these studies. We included them in the revision of the discussion part.
----
Overall, this summary of a single institution's experience with imaging for FUO provides an important summary from a major medical center. There is no "mathematical model", so the title must be changed. Sample size justification should be added. A detailed tabulated comparison of prior published reports and the current study should be added. Several illustrations are suggested for addition, especially the Sheba Medical Center diagnostic workup protocol, study schema, and imaging modality selection criteria. Some minor clarifications or modifications of terminology would be helpful.
We appreciate all of your suggestions. As recommended, we have revised the title to better represent the article. We have also added a sample size justification. We have expanded the number of published reports included that discuss the use of PET/CT in the work-up of FUO with a comparison of their findings, described in the introduction and at greater length within the discussion. The established protocol for PET/CT is included in our methods. However, there is not set protocol for FUO work-up. Because this was a retrospective study and patients were admitted to different wards, there were discrepancies in how the diagnostic work-up was performed and therefore a fixed schema could not be included in the manuscript. We have modified some of the terminology which we believe will provide further clarification.
Reviewer 2 Report
The manuscript “The utility of 18FDG-PET/CT in diagnosing fever of unknown origin: A mathematical Model” describes a method that evaluates the diagnostic yield of 18FDG-PET/CT in the work-up of FUO. The results proved effective in diagnosing almost half the patients, especially in those 34 with anemia and weight loss. The manuscript is reasonably well prepared but could benefit from editing in a number of places. While this reviewer is quite interested in and excited by this work, major revisions, including some new data, are requested before further consideration for publication.
- Although you already cite some articles about FUO, the mechanism between abnormal metabolism and FUO is an important part of this record review. Could you go into a little more detail about it in the introduction? (Line 46-48)
- In the results of the whole article, PET/CT seems to have its place in the diagnosis of FUO. But the conclusion mentioned that clinicians should be aware of the clinical utility of 18FDG-PET/CT and its place in the work-up of FUO following the completion of routine testing. Didn't that cause some conflict on the description with each other? (Line381-382)
- There seem to be other abnormalities in the upper limbs in Figure 1A. Is there any description that can supplement? (Line193-194)
- Figure 7, the place marked by the "infectious causes of FUO" item is blocked by the mark of the plot area. (Line236)
- "Increased BM uptake" of table 2 repeats the description
- The conclusion part mentions "routine testing", can you describe its negative impact and include it in the evaluation index? (Line 382)
- Figure 7, lack of annotations may confuse the reader. Please add how these pie charts are generated below figure 7 title.
Author Response
The manuscript “The utility of 18FDG-PET/CT in diagnosing fever of unknown origin: A mathematical Model” describes a method that evaluates the diagnostic yield of 18FDG-PET/CT in the work-up of FUO. The results proved effective in diagnosing almost half the patients, especially in those 34 with anemia and weight loss. The manuscript is reasonably well prepared but could benefit from editing in a number of places. While this reviewer is quite interested in and excited by this work, major revisions, including some new data, are requested before further consideration for publication.
- Although you already cite some articles about FUO, the mechanism between abnormal metabolism and FUO is an important part of this record review. Could you go into a little more detail about it in the introduction? (Line 46-48).
In accordance with your comment, we added a paragraph regarding the relationship between metabolism in infectious and inflammatory diseases and PET/CT test, in line 49.
- In the results of the whole article, PET/CT seems to have its place in the diagnosis of FUO. But the conclusion mentioned that clinicians should be aware of the clinical utility of 18FDG-PET/CT and its place in the work-up of FUO following the completion of routine testing. Didn't that cause some conflict on the description with each other? (Line381-382).
In this study, PET/CT was used after completing the regular FUO work-up. Accordingly, we think that based on this study, clinicians should be aware at this point in time that PET/CT should follow. Based on this study, we cannot recommend that PET/CT scans be conducted earlier in work-up algorithm. Additional studies are mandated to answer this question.
- There seem to be other abnormalities in the upper limbs in Figure 1A. Is there any description that can supplement? (Line193-194).
We appreciate your noting the slight darkening in the upper limb territories. These changes
most likely show increased bone marrow uptake, an inflammatory reaction expected
in cases of FUO. Because it does not reflect a localized abnormality, we have chosen not
to address it directly in the figure caption.
- Figure 7, the place marked by the "infectious causes of FUO" item is blocked by the mark of the plot area. (Line236).
Thank you for the comment. The figure has been adjusted.
- "Increased BM uptake" of table 2 repeats the description.
We are grateful for your alerting us to what appears like a repetition and may confusing to the reader. We have changed the table heading and added a column to the table to better convey that each of the lines reflects an individual patient.
- The conclusion part mentions "routine testing", can you describe its negative impact and include it in the evaluation index? (Line 382)
The paragraph was amended to be clearer.
- Figure 7, lack of annotations may confuse the reader. Please add how these pie charts are generated below figure 7 title.
A description of the figure was added below.

Reviewer 3 Report
Dear Authors,
This is a well written manuscript that contributes to the literature on the use of PET-CT for evaluation of FUO.
I have a few suggestions that would strengthen the quality of this manuscript.
Please consider adding additional references regarding those who consider PET-CT to be a mandatory part of a FUO work-up and complementary in addition to the studies showing differences in results surrounding the value of PET-CT in your Introductory section.
I and I believe readers will be curious regarding how you ensured that all patients with a FUO who had a PET-CT had the extensive work-up noted in 2.3? Was this a criteria for being allowed to have a PET-CT for FUO indication at your institution? Did you exclude patients who did not have the complete work-up but had a PET-CT as part of a FUO evaluation?
The definition of false positives, false negatives, and true negatives is a little challenging. I’m not sure the PET-CT was truly false positive or false negative in these cases although it did not contribute to the FUO diagnosis. I would suggest finding other wording to define these groups. For example a negative PET in a patient between flares of FMF is not falsely negative, and the diagnosis can be made clinically and with a response to colchicine.
Table 1 is a very important table but I found it confusing. Just looking at the first variable it appears that 80 of 128 patients were male for 62.5% men. I think it would be more helpful to the readers to know what % of men had a contributory PET and what % had a noncontributory PET. A p value for the overall % with a contributory PET for the total 128 could then be calculated versus for the men with a contributory PET. Showing what % of the group for each underlying condition had a contributory PET would be helpful to the reader. The laboratory tests are given as averages, but it might be more interesting to know what percentage of patients had an abnormal test than an average of the results. For example I suspect that a PET-CT will add little to the work-up in a patient with a normal CRP.
If you are able to determine what percentage of patients who were seen at their institution over this time period with a true FUO (meeting the criteria for the study) had a PET-CT it would be helpful to readers to know how much referral bias plays into the results of this study.
Author Response
Dear Authors,
This is a well written manuscript that contributes to the literature on the use of PET-CT for evaluation of FUO.
I have a few suggestions that would strengthen the quality of this manuscript.
Please consider adding additional references regarding those who consider PET-CT to be a mandatory part of a FUO work-up and complementary in addition to the studies showing differences in results surrounding the value of PET-CT in your Introductory section.
In line 62, we added a paragraph about the use of PET/CT earlier in the algorithm, citing the relevant studies.
I and I believe readers will be curious regarding how you ensured that all patients with a FUO who had a PET-CT had the extensive work-up noted in 2.3? Was this a criteria for being allowed to have a PET-CT for FUO indication at your institution? Did you exclude patients who did not have the complete work-up but had a PET-CT as part of a FUO evaluation?
The extensive work-up noted in 2.3 is the accepted basic work-up for every patient who is admitted to the internal medicine departments in our institution. If this work-up does not reveal the fever origin, we proceed to next step which can be CT, PET/CT, bone marrow biopsy or other laboratory/serologic tests which is left to the clinician’s judgement. In our extensive retrospective review of the patients’ data, we did not seek to ensure that every test was done before sending the patients to PET/CT. However, we believe most patients completed a comprehensive work-up. We have added this issue to the limitations section in the discussion in line 394.
The definition of false positives, false negatives, and true negatives is a little challenging. I’m not sure the PET-CT was truly false positive or false negative in these cases although it did not contribute to the FUO diagnosis. I would suggest finding other wording to define these groups. For example a negative PET in a patient between flares of FMF is not falsely negative, and the diagnosis can be made clinically and with a response to colchicine.
We thank you for this comment and looked into it carefully. We agree that these definitions are challenging and not entirely precise. We agree that while in some patients, the positive findings did not lead to the fever origin, but they are signs of a pathological process that might have clinical importance. Nonetheless, after reviewing the relevant literature, we found that this definition is widely used and we believe that it will be better understood by the reader rather than crafting a new definition that is different from what is accepted in the literature.
Table 1 is a very important table but I found it confusing. Just looking at the first variable it appears that 80 of 128 patients were male for 62.5% men. I think it would be more helpful to the readers to know what % of men had a contributory PET and what % had a noncontributory PET. A p value for the overall % with a contributory PET for the total 128 could then be calculated versus for the men with a contributory PET. Showing what % of the group for each underlying condition had a contributory PET would be helpful to the reader. The laboratory tests are given as averages, but it might be more interesting to know what percentage of patients had an abnormal test than an average of the results. For example I suspect that a PET-CT will add little to the work-up in a patient with a normal CRP.
We agree that presenting the data in the way you suggested is highly informative. Nevertheless, we chose to convey it in the way that is most typically used in the literature. Though the way we chose may make it difficult for the reader to understand the distribution of smokers in contributory and non-contributory tests, it has the advantage of presenting data regarding the non-smokers without the need for another line in the chart for non-smokers.
If you are able to determine what percentage of patients who were seen at their institution over this time period with a true FUO (meeting the criteria for the study) had a PET-CT it would be helpful to readers to know how much referral bias plays into the results of this study.
We thank you for this enlightening comment and agree that such data would have been helpful. Unfortunately, this data is not available to us.

Round 2
Reviewer 1 Report
This revised manuscript responds point-by-point to the initial review. Substantial modifications were made. Each issue raised in the initial review was resolved or rebutted with clear justifications.
As a consequence, the revised manuscript is significantly more complete and the information added has strengthened the presentation.
Due to the high clinical significance of the topic and the exceptional quality/reputation of the Sheba Medical Center, this paper should have a high impact.
The only question is that 2 of the suggested references were not included in the bibliography:
FDG-PET/CT in Fever of Unknown Origin, Bacteremia, and Febrile Neutropenia.
Hess S.
PET Clin. 2020 Apr;15(2):175-185. doi: 10.1016/j.cpet.2019.11.002.
Contribution of 18F-FDG PET/CT in a case-mix of fever of unknown origin and inflammation of unknown origin: a meta-analysis.
Kan Y, Wang W, Liu J, Yang J, Wang Z.
Acta Radiol. 2019 Jun;60(6):716-725. doi: 10.1177/0284185118799512.